# Ensemble Learning of Hybrid Acoustic Features for Speech Emotion Recognition

**Kudakwashe Zvarevashe**  **and Oludayo Olugbara** *

ICT and Society Research Group, South Africa Luban Workshop, Durban University of Technology, Durban 4001, South Africa; kudakwashe.zvarevashe@gmail.com

* Correspondence: oludayoo@dut.ac.za

**Abstract:** Automatic recognition of emotion is important for facilitating seamless interactivity between a human being and intelligent robot towards the full realization of a smart society. The methods of signal processing and machine learning are widely applied to recognize human emotions based on features extracted from facial images, video files or speech signals. However, these features were not able to recognize the fear emotion with the same level of precision as other emotions. The authors propose the agglutination of prosodic and spectral features from a group of carefully selected features to realize hybrid acoustic features for improving the task of emotion recognition. Experiments were performed to test the effectiveness of the proposed features extracted from speech files of two public databases and used to train five popular ensemble learning algorithms. Results show that random decision forest ensemble learning of the proposed hybrid acoustic features is highly effective for speech emotion recognition.

**Keywords:** emotion recognition; ensemble algorithm; feature extraction; hybrid feature; machine learning; supervised learning

---

## 1. Introduction

Emotion plays an important role in the daily interpersonal interactions and is considered an essential skill for human communication [1]. It helps humans to understand the opinions of others by conveying feelings and giving feedback to people. Emotion has many useful benefits of affective computing and cognitive activities such as rational decision making, perception and learning [2]. It has opened up an exhilarating research agenda because constructing an intelligent robotic dialogue system that can recognize emotions and precisely respond in the manner of human conversation is presently arduous. The requirement of emotion recognition is steadily increasing with the pervasiveness of intelligent systems [3]. Huawei intelligent video surveillance systems, for instance, can support real-time tracking of a person in a distressed phase through emotion recognition. The capability to recognize human emotions is considered an essential future requirement of intelligent systems that are inherently supposed to interact with people to a certain degree of emotional intelligence [4]. The necessity to develop emotionally intelligent systems is exceptionally important for the modern society of the internet of things (IoT) because such systems have great impact on decision making, social communication and smart connectivity [5].

Practical applications of emotion recognition systems can be found in many domains such as audio/video surveillance [6], web-based learning, commercial applications [7], clinical studies, entertainment [8], banking [9], call centers [10], computer games [11] and psychiatric diagnosis [12]. In addition, other real applications include remote tracking of persons in a distressed phase, communication between human and robots, mining sentiments of sport fans and customer care services [13], where emotion is perpetually expressed. These numerous applications have led to the

development of emotion recognition systems that use facial images, video files or speech signals [14]. In particular, speech signals carry emotional messages during their production [15] and have led to the development of intelligent systems habitually called speech emotion recognition systems [16]. There is an avalanche of intrinsic socioeconomic advantages that make speech signals a good source for affective computing. They are economically easier to acquire than other biological signals like electroencephalogram, electrooculography and electrocardiograms [17], which makes speech emotion recognition research attractive [18]. Machine learning algorithms extract a set of speech features with a variety of transformations to appositely classify emotions into different classes. However, the set of features that one chooses to train the selected learning algorithm is one of the most important tools for developing effective speech emotion recognition systems [3,19]. Research has suggested that features extracted from the speech signal have a great effect on the reliability of speech emotion recognition systems [3,20], but selecting an optimal set of features is challenging [21].

Speech emotion recognition is a difficult task because of several reasons such as an ambiguous definition of emotion [22] and the blurring of separation between different emotions [23]. Researchers are investigating different heterogeneous sources of features to improve the performance of speech emotion recognition systems. In [24], performances of Mel-frequency cepstral coefficient (MFCC), linear predictive cepstral coefficient (LPCC) and perceptual linear prediction (PLP) features were examined for recognizing speech emotion that achieved a maximum accuracy of 91.75% on an acted corpus with PLP features. This accuracy is relatively low when compared to the recognition accuracy of 95.20% obtained for a fusion of audio features based on a combination of MFCC and pitch for recognizing speech emotion [25]. Other researchers have tried to agglutinate different acoustic features with the optimism of boosting accuracy and precision rates of speech emotion recognition [25,26]. This technique has shown some improvement, nevertheless it has yielded low accuracy rates for the fear emotion in comparison with other emotions [25,26]. Semwal et al. [26] fused some acoustic features such as MFCC, energy, zero crossing rate (ZCR) and fundamental frequency that gave an accuracy of 77.00% for the fear emotion. Similarly, Sun et al. [27] used a deep neural network (DNN) to extract bottleneck features that achieved an accuracy of 62.50% for recognizing the fear emotion. The overarching objective of this study was to construct a set of hybrid acoustic features (HAFs) to improve the recognition of the fear emotion and other emotions from speech signal with high precision. This study has contributed to the understanding of the topical theme of speech emotion recognition in the following unique ways.

- The application of a specialized software to extract highly discriminating speech emotion feature representations from multiple sources such as prosodic and spectral to achieve an improved precision in emotion recognition.
- The agglutination of the extracted features using the specialized software to form a set of hybrid acoustic features that can recognize the fear emotion and other emotions from speech signal better than the state-of-the-art unified features.
- The comparison of the proposed set of hybrid acoustic features with other prominent features of the literature using popular machine learning algorithms to demonstrate through experiments, the effectiveness of our proposal over the others.

The content of this paper is succinctly organized as follows. Section 1 provides the introductory message, including the objective and contributions of the study. Section 2 discusses the related studies in chronological order. Section 3 designates the experimental databases and the study methods. The details of the results and concluding statements are given in Sections 4 and 5 respectively.

## 2. Related Studies

Speech emotion recognition research has been exhilarating for a long time and several papers have presented different ways of developing systems for recognizing human emotions. The authors in [3] presented a majority voting technique (MVT) for detecting speech emotion using fast correlation

based feature (FCBF) and Fisher score algorithms for feature selection. They extracted 16 low-level features and tested their methods over several machine learning algorithms, including artificial neural network (ANN), classification and regression tree (CART), support vector machine (SVM) and K-nearest neighbor (KNN) on berlin emotion speech database (Emo-DB). Kerkeni et al. [11] proposed a speech emotion recognition method that extracted MFCC with modulation spectral (MS) features and used recurrent neural network (RNN) learning algorithm to classify seven emotions from Emo-DB and Spanish database. The authors in [15] proposed a speech emotion recognition model where glottis was used for compensation of glottal features. They extracted features of glottal compensation to zero crossings with maximal teager (GCZCMT) energy operator using Taiyuan University of technology (TYUT) speech database and Emo-DB.

　　Evaluation of feature selection algorithms on a combination of linear predictive coefficient (LPC), MFCC and prosodic features with three different multiclass learning algorithms to detect speech emotion was discussed in [19]. Luengo et al. [20] used 324 spectral and 54 prosody features combined with five voice quality features to test their proposed speech emotion recognition method on the Surrey audio-visual expressed emotion (SAVEE) database after applying the minimal redundancy maximal relevance (mRMR) to reduce less discriminating features. In [28], a method for recognizing emotions in an audio conversation based on speech and text was proposed and tested on the SemEval-2007 database using SVM. Liu et al. [29] used the extreme learning machine (ELM) method for feature selection that was applied to 938 features based on a combination of spectral and prosodic features from Emo-DB for speech emotion recognition. The authors in [30] extracted 988 spectral and prosodic features from three different databases using the OpenSmile toolkit with SVM for speech emotion recognition. Stuhlsatz et al. [31] introduced the generalized discriminant analysis (GerDA) based on DNN to recognize emotions from speech using the OpenEar specialized software to extract 6552 acoustic features based on 39 functional of 56 acoustic low-level descriptor (LLD) from Emo-DB and speech under simulated and actual stress (SUSAS) database. Zhang et al. [32] applied the cooperative learning method to recognize speech emotion from FAU Aibo and SUSAS databases using GCZCMT based acoustic features.

　　In [33], Hu moments based weighted spectral features (HuWSF) were extracted from Emo-DB, SAVEE and Chinese academy of sciences - institute of automation (CASIA) databases to classify emotions from speech using SVM. The authors used HuWSF and multicluster feature selection (MCFS) algorithm to reduce feature dimensionality. In [34], extended Geneva minimalistic acoustic parameter set (eGeMAPS) features were used to classify speech emotion, gender and age from the Ryerson audio-visual database of emotional speech and song (RAVDESS) using the multiple layer perceptron (MLP) neural network. Pérez-Espinosa et al. [35] analyzed 6920 acoustic features from the interactive emotion dyadic motion capture (IEMOCAP) database to discover that features based on groups of MFCC, LPC and cochleagrams are important for estimating valence, activation and dominance emotions in speech respectively. The authors in [36] proposed a DNN architecture for extracting informative feature representatives from heterogeneous acoustic feature groups that may contain redundant and unrelated information. The architecture was tested by training the fusion network to jointly learn highly discriminating acoustic feature representations from the IEMOCAP database for speech emotion recognition using SVM to obtain an overall accuracy of 64.0%. In [37], speech features based on MFCC and facial on maximally stable extremal region (MSER) were combined to recognize human emotions through a systematic study on Indian face database (IFD) and Emo-DB.

　　Narendra and Alku [38] proposed a new dysarthric speech classification method from a coded telephone speech using glottal features with a DNN based glottal inverse filtering method. They considered two sets of glottal features based on time and frequency domain parameters plus parameters based on principal component analysis (PCA). Their results showed that a combination of glottal and acoustic features resulted in an improved classification after applying PCA. In [39], four pitch and spectral energy features were combined with two prosodic features to distinguish two high activation states of angry and happy plus low activation states of sadness and boredom for speech

emotion recognition using SVM with Emo-DB. Alshamsi et al. [40] proposed a smart phone method for automated facial expression and speech emotion recognition using SVM with MFCC features extracted from SAVEE database.

Li and Akagi [41] presented a method for recognizing emotions expressed in a multilingual speech using the Fujitsu, Emo-DB, CASIA and SAVEE databases. The highest weighted average precision was obtained after performing speaker normalization and feature selection. The authors in [42] used MFCC related features for recognizing speech emotion based on an improved brain emotional learning (BEL) model inspired by the emotional processing mechanism of the limbic system in human brain. They tested their method on CASIA, SAVEE and FAU Aibo databases using linear discriminant analysis (LDA) and PCA for dimensionality reduction to obtain the highest average accuracy of 90.28% on CASIA database. Mao et al. [43] proposed the emotion-discriminative and domain-invariant feature learning method (EDFLM) for recognizing speech emotion. In their method, both domain divergence and emotion discrimination were considered to learn emotion-discriminative and domain-invariant features using emotion label and domain label constraints. Wang et al. [44] extracted MFCC, Fourier parameters, fundamental frequency, energy and ZCR from three different databases of Emo-DB, Chinese elderly emotion database (EESDB) and CASIA for recognizing speech emotion. Muthusamy et al. [45] extracted a total of 120 wavelet packet energy and entropy features from speech signals and glottal waveforms from Emo-DB, SAVEE and Sahand emotional speech (SES) databases for speech emotion recognition. The extracted features were enhanced using the Gaussian mixture model (GMM) with ELM as the learning algorithm.

Zhu et al. [46] used a combination of acoustic features based on MFCC, pitch, formant, short-term ZCR and short-term energy to recognize speech emotion. They extracted the most discriminating features and performed classification using the deep belief network (DBN) with SVM. Their results showed an average accuracy of 95.8% on the CASIA database across six emotions, which is an improvement when compared to other related studies. Álvarez et al. [47] proposed a classifier subset selection (CSS) for the stacked generalization to recognize speech emotion. They used the estimation of distribution algorithm (EDA) to select optimal features from a collection of features that included eGeMAPS and SVM for classification that achieved an average accuracy of 82.45% on Emo-Db. Bhavan et al. [48] used a combination of MFCCs, spectral centroids and MFCC derivatives of spectral features with a bagged ensemble algorithm based on Gaussian kernel SVM for recognizing speech emotion that achieved an accuracy of 92.45% on Emo-DB. Shegokar et al. [49] proposed the use of continuous wavelet transform (CWT) with prosodic features to recognize speech emotion. They used PCA for feature transformation with quadratic kernel SVM as a classification algorithm that achieved an average accuracy of 60.1% on the RAVDESS database. Kerkeni et al. [50] proposed a model for recognizing speech emotion using empirical mode decomposition (EMD) based on optimal features that included the reconstructed signal based on Mel frequency cepstral coefficient (SMFCC), energy cepstral coefficient (ECC), modulation frequency feature (MFF), modulation spectral (MS) and frequency weighted energy cepstral coefficient (FECC). They achieved an average accuracy of 91.16% on the Spanish database using the RNN algorithm for classification.

Emotion recognition is still a great challenge because of several reasons as previously alluded in the introductory message. Further reasons include the existence of a gap between acoustic features and human emotions [31,36] and the non-existence of a solid theoretical foundation relating the characteristics of voice to the emotions of a speaker [20]. These intrinsic challenges have led to the disagreement in the literature on which features are best for speech emotion recognition [20,36]. The trend in the literature shows that a combination of heterogeneous acoustic features is promising for speech emotion recognition [20,29,30,36,39], but how to effectively unify the different features is highly challenging [21,36]. The importance of selecting relevant features to improve the reliability of speech emotion recognition systems is strongly emphasized in the literature [3,20,29]. Researchers frequently apply specialized software such as OpenEar [31,32,35], OpenSmile [30,32,34,36] and Praat [35] for extraction, selection and unification of speech features from heterogeneous sources to ease the intricacy

inherent in the processes. Moreover, the review of numerous literature has revealed that different learning algorithms are trained and validated with the specific features extracted from public databases for speech emotion recognition. Table 1 shows the result of the comparative analysis of our method with the related methods in terms of the experimental database, type of recording, number of emotions, feature set, classification method and maximum percentage accuracy results obtained. It is conspicuous from the comparative analysis that our emotion recognition method is highly promising because it has achieved the highest average accuracy of 99.55% across two dissimilar public experimental databases.

**Table 1.** A comparison of our method with related methods.

| Reference | Database | Type of Recording | Number of Emotions | Feature Set | Classification Method | Result (%) |
|---|---|---|---|---|---|---|
| [3] | Emo-DB | Acted | Angry, Happy, Neutral, Sad | Energy + MFCC + ZCR + voicing probability + fundamental frequency | FCBF + MVT | 84.19 |
| [11] | Spanish | Acted | Anger, Disgust, Fear, Neutral, Surprise, Sadness, Joy | MFCC + MS | RNN classifier | 90.05 |
| [15] | Emo-DB | Acted | Angry, Happy, Neutral | GCZCMT | SVM | 84.45 |
| [19] | Emo-DB | Acted | Anger, Happiness, Neutral, Sadness | Prosodic + sub-band + MFCC + LPC | SFS algorithm + SVM | 83.00 |
| [28] | eNTERFACE'05 | Elicited | Disgust, Surprise, Happy, Anger, Sad, Fear, | Pitch, energy, formants, intensity and ZCR + text | SVM | 90.00 |
| [29] | CASIA | Acted | Neutral, Happy, Sadness, Fear, Angry, Surprise | Prosodic + quality characteristics + MFCC | correlation analysis + Fisher + ELM decision tree | 89.60 |
| [36] | IEMOCAP | Acted | Angry, Happy, Neutral, Sad | IS10 + MFCCs + eGemaps + SoundNet + VGGish | SVM | 64.00 |
| [39] | Emo-DB | Acted | Anger, Boredom, Happy, Neutral, Sadness | Prosodic features + paralinguistic features | SVM | 94.90 |
| [42] | CASIA | Acted | Surprise, Happy, Sad, Angry, Fear, Neutral | MFCC | GA-BEL + PCA + LDA | 90.28 |
| [46] | CASIA | Acted | Angry, Fear, Happy, Neutral, Surprise, Sad | MFCC, pitch, formant, ZCR and short-term energy | SVM + DBN | 95.80 |
| [47] | Emo-DB | Acted | Sadness, Fear, Joy, Anger, Surprise, Disgust, Neutral | eGeMAPS | CSS stacking system +SVM | 82.45 |
| [48] | Emo-DB | Acted | Anger, Happiness, Sadness, Boredom, Neutral, Disgust, Fear, | MFCCs | bagged ensemble of SVMs | 92.45 |
| [49] | RAVDESS | Acted | Neutral, Surprise, Happy, Angry, Calm, Sad, Fearful, Disgust | CWT, prosodic coefficients | SVM | 60.10 |
| [50] | Spanish | Acted | Anger, Joy, Disgust, Neutral, Surprise, Fear, Sadness | SMFCC, ECC, MFF, MS and EFCC | RNN | 91.16 |
| Proposed model | RAVDESS/SAVEE | Acted | Angry, Sad, Happy, Disgust, Calm, Fear, Neutral, Surprise | Prosodic + spectral | RDF ensemble | 99.55 |

## 3. Materials and Methods

Materials used for this study included speech emotion multimodal databases of RAVDESS [51] and SAVEE [52]. The databases, because of their popularity were chosen to test for the effectiveness

of the proposed HAF against two other sets of features. The study method followed three phases of feature extraction, feature selection and classification, which were subsequently described. The method was heralded by the feature extraction process that involves the abstraction of prosodic and spectral features from the raw audio files of RAVDESS and SAVEE databases. This phase was subsequently followed by the feature selection process that involved the filtration, collection and agglutination of features that have high discriminating power of recognizing emotions in human speech. Classification was the last phase that involved the application of the selected learning algorithm to recognize human emotions and a comparative analysis of experimental results of classification. In the classification phase, several experiments were carried out on a computer with an i7 2.3GHz processor and 8 GB of random access memory (RAM). The purpose of the experiments was to apply standard metrics of accuracy, precision, recall and F1-score to evaluate the effectiveness of the proposed HAF with respect to a given database and ensemble learning algorithm.

*3.1. Databases*

The speech emotion database is an important precursor for analyzing speech and recognizing emotions. The database provides speech data for training and testing the effectiveness of emotion recognition algorithms [15]. RAVDESS and SAVEE are two public speech emotion databases used in this study for experiments to verify the effectiveness of the proposed HAF for emotion recognition. The two selected experimental databases were fleetingly discussed in this subsection of the paper.

3.1.1. RAVDESS

RAVDESS is a gender balanced set of validated speeches and songs that consists of eight emotions of 24 professional actors speaking similar statements in a North American accent. It is a multiclass database of angry, calm, disgust, fear, happy, neutral, sad and surprise emotions with 1432 American English utterances. Each of the 24 recorded vocal utterances comprises of three formats, which are audio-only (16bit, 48kHz .wav), audio-video (720p H.264, AAC 48kHz, .mp4) and video-only (no sound). The audio-only files were used across all the eight emotions because this study concerns speech emotion recognition. Figure 1 shows that angry, calm, disgust, fear, happy and sad emotion classes constituted 192 audio files each. The surprise emotion had 184 files and the neutral emotion had the lowest number of audio files of 96.

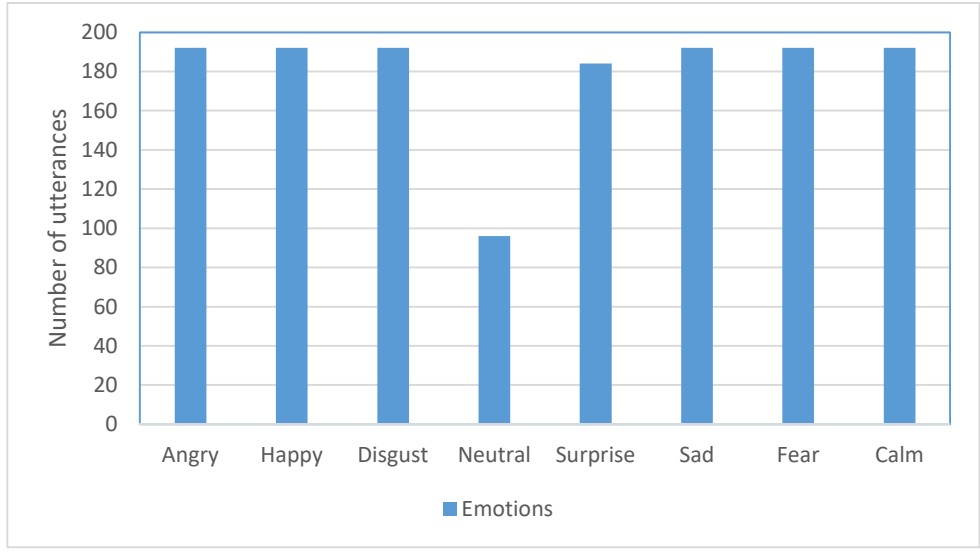

**Figure 1.** The Ryerson audio-visual database of emotional speech and song (RAVDESS).

### 3.1.2. SAVEE

SAVEE is a speech emotion database that consists of recordings from four male actors in seven different emotion classes. The database comprises of a total of 480 British English utterances, which is quite different from the North American accent used in the RAVDESS database. These vocal utterances were processed and labeled in a standard media laboratory with high quality audio-visual equipment. The recorded vocal utterances were classified into seven classes of angry, disgust, fear, happy, neutral, sad and surprise emotional expressions. The neutral emotion constituted 120 audio files, while all the other remaining emotions comprised of 60 audio files each as illustrated in Figure 2.

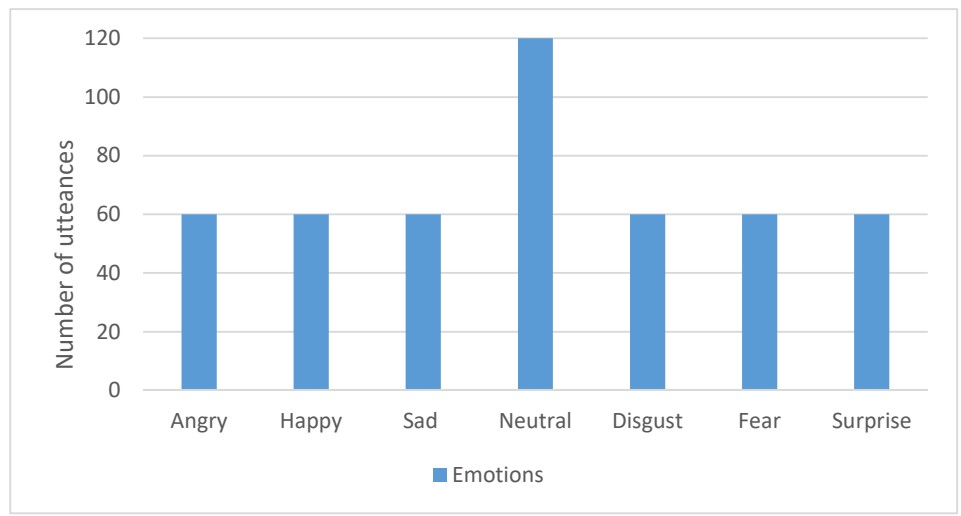

**Figure 2.** The Surrey audio-visual expressed emotion (SAVEE).

### 3.2. Feature Extraction

The speech signal carries a large number of useful information that reflects emotion characteristics such as gender, age, stuttering and identity of the speaker. Feature extraction is an important mechanism in audio processing to capture the information in a speech signal and most of the related studies have emphasized on the extraction of low-level acoustic features for speech recognition. Feature representation plays a prominent role in distinguishing the speech of speakers from each other. Since no universal consensus on the best features for speech recognition, certain authors have mentioned that prosody carries most of the useful information about emotions and can create a suitable feature representation [20]. Moreover, spectral features describe the properties of a speech signal in the frequency domain to identify important aspects of speech and manifest the correlation between vocal movements and changes of channel shape [35,42]. Prosodic and spectral acoustic features were suggested as the most important characteristics of speech because of the improved recognition results of their integration [25,42,53,54]. The blend of prosodic and spectral features is capable of enhancing the performance of emotion recognition, giving better recognition accuracy and a lesser number of emotions in a gender dependent model than many existing systems with the same number of emotions [25]. Banse et al. [54] examined vocal cues for 14 emotions using a combination of prosodic speech features and spectral information in a voiced segment and unvoiced segment to achieve impressive results.

The present authors have applied the selective approach [35] to extract prosodic and spectral features, taking cognizance of features that could be useful for improving emotion recognition. These are features that have produced positive results in the related studies and those from the other comparable tasks. We carefully applied the specialized jAudio software [55] to construct MFCC1, MFCC2 and HAF as three sets of features representing several voice aspects. MFCC1 is the set of MFCC features inspired by the applications of MFCC features [24,40,42]. MFCC2 is the set of features based on

MFCC, energy, ZCR and fundamental frequency as inspired by the fusion of MFCC with other acoustic features [11,25,26,44,46]. The HAF is the proposed set of hybrid acoustic features of prosodic and spectral carefully selected based on the interesting results in the literature [11,20,24–26,29,30,39,40,42,44,46,48].

The feature extraction process was aided by the application of a specialized jAudio to effectively manage the inherent complexity in the selection process. The jAudio software is a digital signal processing library of advanced audio feature extraction algorithms designed to eliminate effort duplication in manually calculating emotion features from an audio signal. It provides a unique method of handling multidimensional features, dependency and allows for iterative development of new features from the existing ones through the meta-feature template. Moreover, it simplifies the intrinsic stiffness of the existing feature extraction methods by allowing low-level features to be fused to build increasingly high-level musically meaningful features. In addition, it provides audio samples as simple vectors of features and offers aggregation functions that aggregate a sequence of separate feature sets into a single feature vector.

### 3.2.1. Prosodic Features

Prosodic features are acoustic features prominently used in emotion recognition and speech signal processing because they carry essential paralinguistic information. This information complements a message with an intention that can paint a flawless picture about attitude or emotion [35,56]. In addition, prosodic features are considered as suprasegmental information because they help in defining and structuring the flow of speech [35]. Prosodic continuous speech features such as pitch and energy convey much content of emotions in speech [57] and are important for delivering the emotional cues of the speakers. These features include formant, timing and articulation features and they characterize the perceptual properties of speech typically used by human beings to perform different speech tasks [57].

The present authors have included three important prosodic features of energy, fundamental frequency and ZCR. Signal energy models the voice intensity, volume or loudness and reflects the pause and ascent of the voice signal. It is often associated with the human respiratory system and is one of the most important characteristics of human aural perception. The logarithm function is often used to reflect minor changes of energy because the energy of an audio signal is influenced by the recording conditions. Fundamental frequency provides tonal plus rhythmic characteristics of a speech and carries useful information about the speaker. ZCR determines the information about the number of times a signal waveform crosses the zero amplitude line because of a transition from a positive/negative value to a negative/positive value in a given time. It is suitable for detecting voice activity, end point, voiced sound segment, unvoiced sound segment, silent segment and approximating the measure of noisiness in speech [58]. ZCR is an acoustic feature that has been classified as a prosodic feature [49,59,60]. In particular, energy and pitch were declared prosody features with a low frequency domain while ZCR and formants are high frequency features [60].

### 3.2.2. Spectral Features

Spectral features of this study include timbral features that have been successful in music recognition [61]. Timbral features define the quality of a sound [62] and they are a complete opposite of most general features like pitch and intensity. It has been revealed that a strong relationship exists between voice quality and emotional content in a speech [58]. Siedenburg et al. [63] considered spectral features as significant in distinguishing between classes of speech and music. They claimed that the temporal evolution of the spectrum of audio signals mainly accounts for the timbral perception. Timbral features of a sound help in differentiating between sounds that have the same pitch and loudness and they assist in the classification of audio samples with similar timbral features into unique classes [61]. The application of spectral timbral features like spectral centroid, spectral roll-off point, spectral flux, time domain zero crossings and root mean squared energy amongst others has been demonstrated for speech analysis [64].

The authors have included eight spectral features of MFCC, spectral roll-off point, spectral flux, spectral centroid, spectral compactness, fast Fourier transforms, spectral variability and LPCC. MFCC is one of the most widely used feature extraction methods for speech analysis because of its computational simplicity, superior ability of distinction and high robustness to noise [42]. It has been successfully applied to discriminate phonemes and can determine what is said and how it is said [35]. The method is based on the human hearing system that provides a natural and real reference for speech recognition [46]. In addition, it is based on the sense that human ears perceive sound waves of different frequencies in a non-linear mode [42]. Spectral roll-off point defines a frequency below which at least 85% of the total spectral energy are present and provides a measure of spectral shape [65]. Spectral flux characterizes the dynamic variation of spectral information, is related to perception of music rhythm and it captures spectrum difference between two adjacent frames [66].

Spectral centroid describes the center of gravity of the magnitude spectrum of short-time Fourier transform (STFT). It is a spectral moment that is helpful in modeling sharpness or brightness of sound [67]. Spectral compactness is obtained by comparing the components in the magnitude spectrum of a frame and magnitude spectrum of neighboring frames [68]. Fast Fourier transform (FFT) is a useful method of analyzing the frequency spectrum of a speech signal and features based on the FTT algorithm have the strongest frequency component in Hertz [69,70]. Spectral variability is a measure of variance of the magnitude spectrum of a signal and is attributed to different sources, including phonetic content, speaker, channel, coarticulation and context [71]. Spectral compactness is an audio feature that measures the noisiness of a speech signal and is closely related to the spectral smoothness of speech signal [72]. LPCCs are spectral features obtained from the envelope of LPC and are computed from sample points of a speech waveform. They have low vulnerability to noise, yielded a lower error rate in comparison to LPC features and can discriminate between different emotions [73]. LPCC features are robust, but they are not based on an auditory perceptual frequency scale like MFCC [74].

*3.3. Feature Selection*

Feature selection is a process of filtering insignificant features to create a subset of the most discriminating features in the input data. It reduces the cause of dimensionality and improves the success of a recognition algorithm. Based on the literature review, we observed that most methods used for feature selection are computationally expensive because of the enormous processing time involved. Moreover, accuracy and precision values of some of these methods are reduced because of redundancy, less discriminating features and poor accuracy values in certain emotions such as the fear emotion. The authors have selected 133 MFCC spectral features for the MFCC1 and 90 features based on MFCC, ZCR, energy and fundamental frequency for the MFCC2. In a nutshell, the effectiveness of the HAF was tested against one homogenous set of acoustic features (MFCC1) and another hybrid set of acoustic features (MFCC2).

Table 2 presents the list of prosodic and spectral features of this study in terms of group, type and number of features with their statistical characteristics obtained with the aid of jAudio that generated a csv file with numerical values for each feature together with the respective emotion class. The literature inspired brute force approach was then explored to select non-redundant features that are likely to help improve emotion recognition performance. This implies inspecting through a series of experiments for possible combinations of features to discover the right features that define HAF. This approach is although laborious to require the help of a stochastic optimization technique, it has yielded excellent performance results in this study.

**Table 2.** List of the selected prosodic and spectral features.

| Group | Type | Number |
|---|---|---|
| **Prosodic** | | |
| Energy | Logarithm of Energy | 10 |
| Pitch | Fundamental Frequency | 70 |
| Times | Zero Crossing Rate | 24 |
| **Spectral** | | |
| Cepstral | MFCC | 133 |
| Shape | Spectral Roll-off Point | 12 |
| Amplitude | Spectral Flux | 12 |
| Moment | Spectral Centroid | 22 |
| Audio | Spectral Compactness | 10 |
| Frequency | Fast Fourier Transform | 9 |
| Signature | Spectral Variability | 21 |
| Envelope | LPCC | 81 |
| **Total** | | **404** |

*3.4. Classification*

Emotion classification is aimed at obtaining an emotional state for the input feature vector using a machine learning algorithm. In this study, classification experiments were conducted using ensemble learning algorithms to verify the effectiveness of the proposed HAF. Ensemble learning is considered a state-of-the-art approach for solving different machine learning problems by combining the predictions of multiple base learners [75]. It improves the overall predictive performance, decreases the risk of obtaining a local minimum and provides a better fit to the data space by combining the predictions of several weak learners into a strong learning algorithm. In this study, we applied bagging and boosting machine learning because they are widely used effective approaches for constructing ensemble learning algorithms [76–79]. Bagging is a technique that utilizes bootstrap sampling to reduce the variance of a decision tree and improve the accuracy of a learning algorithm by creating a collection of learning algorithms that are learned in parallel [79–81]. It performs random sampling with replacement over a simple averaging of all the predictions from different decision trees to give a more robust result than a single decision tree. Boosting creates a collection of learning algorithms that are learned sequentially with early learners to fit decision trees to the data and analyze data for errors. It performs random sampling with replacement over a weighted average and reduces classification bias and variance of a decision tree [79]. One of the bagging ensemble algorithms investigated in this study was random decision forest (RDF), which is an extension over bagging that is popularly called random forest [75,82]. RDF can be constituted by making use of bagging based on the CART approach to raise trees [83]. The other bagging ensemble learning algorithms investigated in this study were Bagging with SVM as the base learner (BSVM) and bagging with the multilayer perceptron neural network as the base learner (BMLP). The boosting ensemble algorithms investigated in this study were the gradient boosting machine (GBM), which extends boosting by combining the gradient descent optimization algorithm with boosting technique [75,82,84], and AdaBoost with CART as the base learner (ABC), which is one of the most widely used boosting algorithm to reduce sensitivity to class label noise [79,85]. AdaBoost is an iterative learning algorithm for constructing a strong classifier by enhancing weak classification algorithms and it can improve data classification ability by reducing both bias and variance through continuous learning [81].

The standard metrics of accuracy, precision, recall and F1-score have been engaged to measure the performances of the learning algorithms with respect to a particular set of features. The accuracy

of a classification algorithm is often judged as one of the most intuitive performance measures. It is the ratio of the number of instances correctly recognized to the total number of instances. Precision is the ratio of the number of positive instances correctly recognized to the total number of positive instances. Recall is the ratio of the number of positive instances correctly recognized to the number of all the instances of the actual class [86]. F1-score constitutes a harmonic mean of precision and recall [87]. Moreover, the training times of the learning algorithms are also compared to analyze the computational complexities of training different ensemble learning classifiers. The flow chart for different configurations of the methods of experimentation is illustrated in Figure 3.

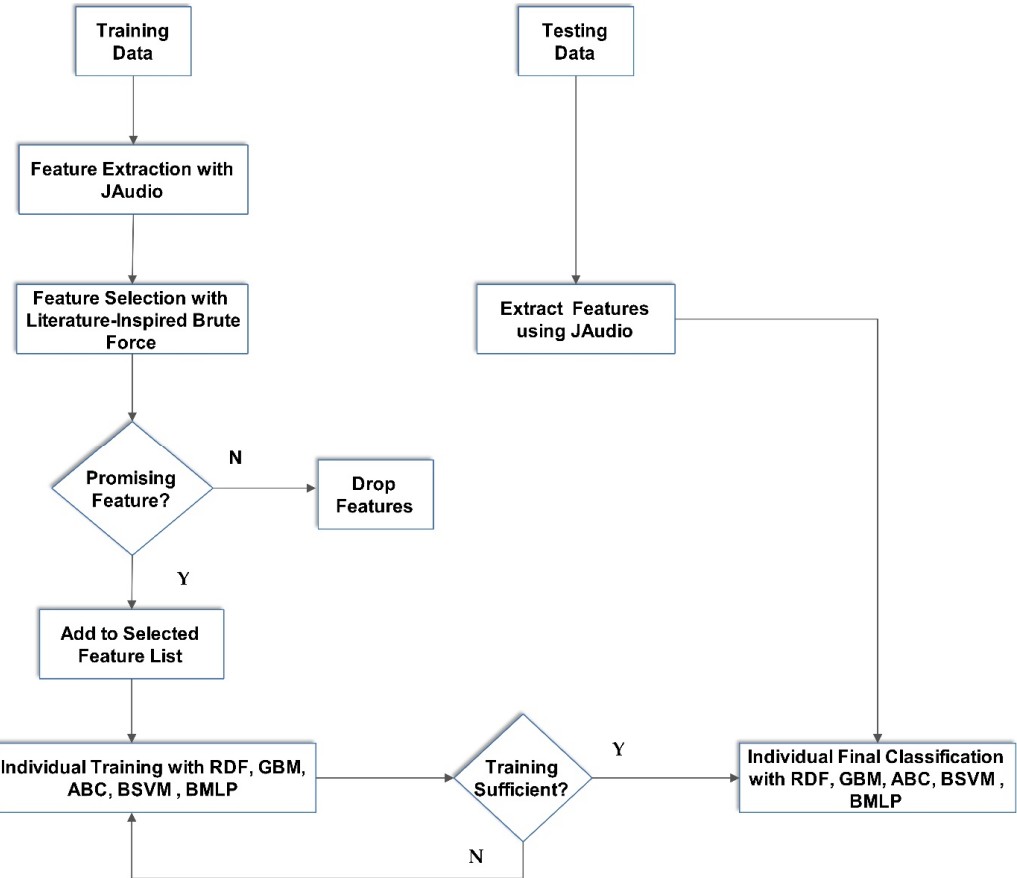

**Figure 3.** Flowchart for different configurations of emotion recognition methods.

## 4. Results and Discussion

The present authors have extracted the set of 404 HAFs from RAVDESS and SAVEE experimental databases. The features were used to train five famous ensemble learning algorithms in order to determine their effectiveness using four standard performance metrics. The performance metrics were selected because the experimental databases were not equally distributed. Data were subjected to 10 fold cross-validation and divided into training and testing groups as the norm permits. Gaussian noise was added to the training data to provide a regularizing effect, reduce overfitting, increase resilience and improve generalization performance of the ensemble learning algorithms.

Table 3 shows the result of the CPU computational time of training an individual ensemble learning algorithm with three different sets of features across two experimental databases. The training time analysis shows that RDF spent the least amount of training time when compared to other ensemble learning algorithms. The least training time was recorded with respect to MFCC1 and MFCC2 irrespective of databases. However, the training time of RDF (0.43) was slightly higher than that of BSVM (0.39) with respect to SAVEE HAF, but it claimed superiority with respect to the

RAVDESS HAF. The reason for this inconsistent result might be because of the smaller data size (480) of SAVEE when compared to the size (1432) of RAVDESS. The experimental results of a study that was aimed at fully assessing the predictive performances of SVM and SVM ensembles over small and large scale datasets of breast cancer show that SVM ensemble based on BSVM method can be the better choice for a small scale dataset [78]. Moreover, it can be observed that GBM and BMLP were generally time consuming when compared to the other ensemble learning algorithms because they took 1172.3 milliseconds to fit MFCC2 RAVDESS and 1319.39 milliseconds to fit MFCC2 SAVEE features respectively.

**Table 3.** Processing time of learning algorithms trained with Mel-frequency cepstral coefficient 1 (MFCC1), Mel-frequency cepstral coefficient 2 (MFCC2) and a hybrid acoustic feature (HAF).

| Classifier | SAVEE Feature | | | RAVDESS Feature | | |
| --- | --- | --- | --- | --- | --- | --- |
| | MFCC1 | MFCC2 | HAF | MFCC1 | MFCC2 | HAF |
| RDF | 0.03 | 0.38 | 0.43 | 0.052 | 1.62 | 0.06 |
| GBM | 2.30 | 579.40 | 343.10 | 5.90 | 1172.30 | 6.90 |
| ABC | 0.47 | 22.16 | 1.38 | 2.59 | 2.79 | 1.89 |
| BSVM | 0.20 | 49.79 | 0.39 | 0.42 | 211.54 | 0.48 |
| BMLP | 3.13 | 1319.39 | 952.46 | 76.00 | 231.16 | 7.18 |

The experimental results of the overall recognition performance obtained are given in Table 4. The percentage average recognition performance rates differ depending on critical factors such as the type of database, the accent of the speakers, the way data were collected, the type of features and the learning algorithm used. In particular, RDF consistently recorded the highest average accuracy, while BMLP consistently recorded the lowest average accuracy across the sets of features, irrespective of databases. These results indicate that RDF gave the highest performance because the computed values of the precision, recall, F1-score and accuracy of the ensemble learning algorithm are generally impressive when compared to other ensemble learning algorithms. However, average recognition accuracies of 61.5% and 66.7% obtained by RDF for the SAVEE MFCC1 and MFCC2 were respectively low, but the values were better than that of BMLP (55.4% and 60.7%) for SAVEE MFCC1 and MFCC2 respectively. In general, an improvement can be observed when a combination of features (MFCC2) was used in an experiment drawing inspiration from the work done in [46], where MFCC features were combined with pitch and ZCR features. In addition, we borrowed the same line of thought from Sarker and Alam [3] and Bhaskar et al. [28] where some of the above mentioned features were combined to form hybrid features.

**Table 4.** Percentage average precision, recall, F1-score and accuracy with confidence intervals of learning algorithms trained with MFCC1, MFCC2 and HAF.

| Classifier/Measure | SAVEE Feature | | | RAVDESS Feature | | |
| --- | --- | --- | --- | --- | --- | --- |
| | MFCC1 | MFCC2 | HAF | MFCC1 | MFCC2 | HAF |
| **RDF** | | | | | | |
| Precision | 63.4 (±0.044) | 78.1 (±0.037) | 99.1(±0.009) | 90.1 (±0.016) | 93.1 (±0.013) | 99.6 (±0.003) |
| Recall | 72.1 (±0.041) | 77.4 (±0.038) | 99.1(±0.009) | 88.9 (±0.016) | 94.0 (±0.012) | 99.5 (±0.004) |
| F1-score | 63.6 (±0.043) | 76.6 (±0.039) | 99.1(±0.009) | 88.5 (±0.017) | 92.5 (±0.014) | 99.8 (±0.002) |
| Accuracy | 61.5 (±0.044) | 66.7 (±0.043) | 99.3 (±0.008) | 90.7 (±0.015) | 93.7 (±0.013) | 99.8 (±0.002) |
| **GBM** | | | | | | |
| Precision | 64.0 (±0.043) | 75.6 (±0.039) | 99.4 (±0.007) | 85.0 (±0.019) | 86.5 (±0.018) | 95.3(±0.011) |
| Recall | 66.6 (±0.043) | 74.6 (±0.039) | 99.1 (±0.009) | 82.9 (±0.020) | 88.1 (±0.017) | 94.8(±0.012) |
| F1-score | 62.6 (±0.044) | 72.7 (±0.040) | 99.3 (±0.008) | 82.6 (±0.020) | 85.9 (±0.018) | 96.8(±0.009) |
| Accuracy | 61.5 (±0.044) | 65.5 (±0.043) | 99.3 (±0.008) | 85.4(±0.018) | 86.7(±0.018) | 92.6(±0.014) |
| **ABC** | | | | | | |
| Precision | 62.9 (±0.044) | 73.4 (±0.040) | 99.1 (±0.009) | 81.3 (±0.020) | 85.4 (±0.018) | 94.3(±0.012) |
| Recall | 64.1 (±0.044) | 73.7 (±0.040) | 99.4 (±0.007) | 84.0 (±0.019) | 87.5 (±0.017) | 94.5(±0.012) |
| F1-score | 63.1 (±0.044) | 74.1 (±0.040) | 97.9 (±0.013) | 82.6 (±0.020) | 84.5 (±0.019) | 95.9(±0.010) |
| Accuracy | 58.0 (±0.044) | 62.8 (±0.043) | 98.0 (±0.008) | 83.0 (±0.020) | 85.2 (±0.018) | 92.0 (±0.014) |
| **BSVM** | | | | | | |
| Precision | 61.4 (±0.044) | 71.4 (±0.041) | 98.7 (±0.010) | 81.0 (±0.020) | 84.1 (±0.019) | 91.1(±0.015) |
| Recall | 61.0 (±0.044) | 72.6 (±0.041) | 99.0 (±0.009) | 82.5 (±0.020) | 82.3(±0.020) | 92.0(±0.014) |
| F1-score | 61.9 (±0.044) | 72.0 (±0.041) | 99.3 (±0.008) | 80.5 (±0.021) | 83.5 (±0.019) | 90.3(±0.015) |
| Accuracy | 56.0 (±0.045) | 61.5 (±0.044) | 96.0 (±0.013) | 82.7 (±0.020) | 84.5 (±0.019) | 91.8 (±0.014) |
| **BMLP** | | | | | | |
| Precision | 59.4 (±0.044) | 69.0 (±0.042) | 97.9 (±0.013) | 78.8 (±0.021) | 77.5 (±0.022) | 93.0 (±0.013) |
| Recall | 60.1 (±0.044) | 72.0 (±0.041) | 98.4 (±0.011) | 75.8 (±0.022) | 80.3 (±0.021) | 88.0(±0.017) |
| F1-score | 59.6 (±0.044) | 70.3 (±0.041) | 98.1 (±0.012) | 74.4(±0.023) | 78.3 (±0.021) | 89.0 (±0.016) |
| Accuracy | 55.4 (±0.045) | 60.7 (±0.044) | 94.6 (±0.002) | 75.6 (±0.022) | 79.3 (±0.021) | 91.3 (±0.015) |

The results in Table 4 can be used to demonstrate the effectiveness of the proposed HAF. The values of the precision, recall, F1-score and accuracy performance measures can be seen to be generally high for the five ensemble learning algorithms using the HAF irrespective of databases. Specifically, RDF and GBM concomitantly recorded the highest accuracy of 99.3% for the SAVEE HAF, but RDF claims superiority with 99.8% accuracy when compared to the 92.6% accuracy for the HAF RAVDESS. Consequently, the results of RDF learning of the proposed HAF were exceptionally inspiring with an average accuracy of 99.55% across the two databases. The ranking of the investigated ensemble learning algorithms in terms of percentage average accuracy computed for HAF across the databases is RDF (99.55%), GBM (95.95%), ABC (95.00%), BSVM (93.90%) and BMLP (92.95%). It can be inferred that application of ensemble learning of HAF is highly promising for recognizing emotions in human speech. In general, high accuracy values computed by all learning algorithms across databases indicate the effectiveness of the proposed HAF for speech emotion recognition. Moreover, the confidence factors show that the true classification accuracies of all the ensemble learning algorithms investigated lie in the range between 0.002% and 0.045% across all the corpora. The results in Table 4 show that HAF significantly improved the performance of emotion recognition when compared to the methods of the SAVEE MFCC related features [42] and RAVDESS wavelet transform features [49] that recorded the highest recognition accuracies of 76.40% and 60.10% respectively.

In particular, the results of precision, recall, F1-score and accuracy per each emotion, learning algorithm and database were subsequently discussed. Table 5 shows that angry, calm and disgust emotions were easier to recognize using RDF and GBM ensemble learning of RAVDESS MFCC1 features. The results were particularly impressive with RDF ensemble learning with perfect precision, recall and F1-score computed by the algorithm, which also recorded highest accuracy values across all emotions. However, BMLP generally gave the least performing result across all emotions. RAVDESS MFCC1 features achieved a precision rate of 92.0% for the fear emotion using RDF ensemble learning, which is relatively high when compared to the results in [27].

**Table 5.** Percentage precision, recall and F1-score and accuracy on RAVDESS MFCC1.

| | Emotion | | | | | | | |
|---|---|---|---|---|---|---|---|---|
| Classifier/Measure | Angry | Calm | Disgust | Fear | Happy | Neutral | Sad | Surprise |
| **RDF** | | | | | | | | |
| Precision | 100.0 | 100.0 | 100.0 | 92.0 | 80.0 | 53.0 | 87.0 | 96.0 |
| Recall | 100.0 | 100.0 | 100.0 | 83.0 | 82.0 | 86.0 | 79.0 | 91.0 |
| F1-score | 100.0 | 100.0 | 100.0 | 87.0 | 81.0 | 66.0 | 83.0 | 94.0 |
| Accuracy | 91.0 | 90.0 | 91.0 | 89.0 | 91.0 | 89.0 | 91.0 | 93.0 |
| **GBM** | | | | | | | | |
| Precision | 94.0 | 100.0 | 100.0 | 87.0 | 69.0 | 39.0 | 87.0 | 85.0 |
| Recall | 100.0 | 100.0 | 100.0 | 74.0 | 69.0 | 78.0 | 72.0 | 87.0 |
| F1-score | 97.0 | 100.0 | 100.0 | 80.0 | 69.0 | 52.0 | 79.0 | 86.0 |
| Accuracy | 86.0 | 85.2 | 87.0 | 81.0 | 83.0 | 85.0 | 86.0 | 90.0 |
| **ABC** | | | | | | | | |
| Precision | 91.0 | 82.0 | 78.0 | 79.0 | 81.0 | 79.0 | 81.0 | 81.0 |
| Recall | 100.0 | 100.0 | 100.0 | 81.0 | 56.0 | 85.0 | 71.0 | 82.0 |
| F1-score | 95.0 | 100.0 | 91.0 | 65.0 | 69.0 | 70.0 | 81.0 | 79.0 |
| Accuracy | 85.0 | 84.0 | 85.0 | 75.0 | 81.0 | 79.0 | 87.0 | 88.0 |
| **BSVM** | | | | | | | | |
| Precision | 83.0 | 84.0 | 82.0 | 75.0 | 75.0 | 77.0 | 79.0 | 89.0 |
| Recall | 82.0 | 83.0 | 91.0 | 73.0 | 79.0 | 81.0 | 84.0 | 87.0 |
| F1-score | 81.0 | 82.0 | 79.0 | 79.0 | 81.0 | 79.0 | 82.0 | 84.0 |
| Accuracy | 83.0 | 85.0 | 81.0 | 79.0 | 80.6 | 81.0 | 86.0 | 86.0 |
| **BMLP** | | | | | | | | |
| Precision | 72.0 | 81.0 | 78.0 | 68.0 | 70.0 | 70.0 | 75.0 | 81.0 |
| Recall | 78.0 | 84.0 | 82.0 | 70.0 | 76.0 | 79.0 | 78.0 | 83.0 |
| F1-score | 74.0 | 82.0 | 80.0 | 67.0 | 71.0 | 75.0 | 77.0 | 80.0 |
| Accuracy | 80.0 | 77.0 | 76.0 | 71.0 | 74.0 | 75.0 | 75.0 | 77.0 |

Table 6 shows the results of percentage precision, recall, F1-score and accuracy performance analysis of RAVDESS MFCC2 features. The performance values were generally higher for recognizing angry, calm and disgust emotions using RDF. However, the performance values were relatively low for happy and neutral emotions showing that the hybridization of MFCC, ZCR, energy and fundamental frequency features yielded a relatively poor result in recognizing happy and neutral emotions using ensemble learning. The MFCC2 combination of acoustic features achieved an improved recognition performance of the fear (90.0%) emotion using RDF in comparison with the report in [44], where 81.0% recognition was achieved for the fear emotion using the CASIA database.

**Table 6.** Percentage precision, recall and F1-score and accuracy on RAVDESS MFCC2.

| Classifier/Measure | Angry | Calm | Disgust | Fear | Happy | Neutral | Sad | Surprise |
|---|---|---|---|---|---|---|---|---|
| | | | | Emotion | | | | |
| **RDF** | | | | | | | | |
| Precision | 100.0 | 100.0 | 100.0 | 91.0 | 95.0 | 64.0 | 90.0 | 100.0 |
| Recall | 100.0 | 100.0 | 100.0 | 89.0 | 78.0 | 100.0 | 90.0 | 95.0 |
| F1-score | 100.0 | 100.0 | 100.0 | 93.0 | 86.0 | 78.0 | 90.0 | 98.0 |
| Accuracy | 96.0 | 93.0 | 92.0 | 90.0 | 94.0 | 91.0 | 96.0 | 97.6 |
| **GBM** | | | | | | | | |
| Precision | 100.0 | 100.0 | 100.0 | 87.0 | 79.0 | 55.0 | 80.0 | 86.0 |
| Recall | 100.0 | 100.0 | 100.0 | 87.0 | 75.0 | 86.0 | 67.0 | 90.0 |
| F1-score | 100.0 | 100.0 | 100.0 | 87.0 | 77.0 | 67.0 | 73.0 | 88.0 |
| Accuracy | 88.0 | 84.0 | 83.0 | 82.0 | 85.0 | 85.0 | 88.0 | 98.6 |
| **ABC** | | | | | | | | |
| Precision | 94.0 | 93.0 | 91.0 | 78.0 | 71.0 | 79.0 | 81.0 | 89.0 |
| Recall | 96.0 | 100.0 | 100.0 | 86.0 | 76.0 | 73.0 | 83.0 | 86.0 |
| F1-score | 92.0 | 95.0 | 100.0 | 79.0 | 74.0 | 76.0 | 81.0 | 86.0 |
| Accuracy | 86.0 | 82.0 | 83.0 | 81.0 | 87.0 | 82.0 | 88.0 | 92.6 |
| **BSVM** | | | | | | | | |
| Precision | 81.0 | 94.0 | 91.0 | 77.0 | 74.0 | 79.0 | 78.0 | 94.0 |
| Recall | 92.0 | 89.0 | 90.0 | 76.0 | 74.0 | 77.0 | 74.0 | 90.0 |
| F1-score | 94.0 | 91.0 | 91.0 | 77.0 | 74.0 | 78.0 | 76.0 | 92.0 |
| Accuracy | 84.0 | 81.0 | 80.0 | 78.0 | 85.0 | 85.0 | 87.0 | 96.0 |
| **BMLP** | | | | | | | | |
| Precision | 76.0 | 89.0 | 86.0 | 72.0 | 70.0 | 76.0 | 74.0 | 83.0 |
| Recall | 75.0 | 86.0 | 86.0 | 74.0 | 67.0 | 75.0 | 72.0 | 85.0 |
| F1-score | 77.0 | 91.0 | 87.0 | 74.0 | 73.0 | 76.0 | 74.0 | 90.0 |
| Accuracy | 85.0 | 81.0 | 78.0 | 73.0 | 78.0 | 77.0 | 78.0 | 84.0 |

Emotion recognition results shown in Table 7 with respect to the RAVDESS HAF were generally impressive across emotions, irrespective of the ensemble learning algorithms utilized. RDF was highly effective in recognizing all the eight emotions because it had achieved perfect accuracy value by 100.0% on almost all the emotions, except fear (98.0%). Moreover, perfect precision and accuracy values were achieved in recognizing the neutral emotion, which was really difficult to identify using other sets of features. In addition, 98.0% accuracy and perfect values obtained for precision, recall and F1-score for the fear emotion is impressive because of the difficulty of recognizing the fear emotion as reported in the literature [25–27]. All the other learning algorithms performed relatively well, considering the fact that noise was added to the training data. These results attest to the effectiveness of the proposed HAF for recognizing emotions in human speech. In addition, the results point to the significance of sourcing highly discriminating features for recognizing human emotions. They further show that the fear emotion could be recognized efficiently when compared to other techniques used in [44]. It can be seen that the recognition of fear emotion was successfully improved using the proposed HAF. In addition, HAF gave better performance than the bottleneck features used for six emotions from CASIA database [27] that achieved a recognition rate of 60.5% for the fear emotion.

**Table 7.** Percentage precision, recall and F1-score and accuracy on RAVDESS HAF.

| Classifier/Measure | Emotion | | | | | | | |
|---|---|---|---|---|---|---|---|---|
| | **Angry** | **Calm** | **Disgust** | **Fear** | **Happy** | **Neutral** | **Sad** | **Surprise** |
| **RDF** | | | | | | | | |
| Precision | 100.0 | 100.0 | 100.0 | 100.0 | 100.0 | 100.0 | 100.0 | 98.0 |
| Recall | 100.0 | 100.0 | 100.0 | 100.0 | 100.0 | 96.0 | 100.0 | 100.0 |
| F1-score | 100.0 | 100.0 | 100.0 | 100.0 | 100.0 | 98.0 | 100.0 | 99.0 |
| Accuracy | 100.0 | 100.0 | 100.0 | 98.0 | 100.0 | 100.0 | 100.0 | 100.0 |
| **GBM** | | | | | | | | |
| Precision | 93.0 | 100.0 | 100.0 | 94.0 | 100.0 | 95.0 | 99.0 | 93.0 |
| Recall | 92.0 | 100.0 | 100.0 | 93.0 | 97.0 | 100.0 | 76.0 | 100.0 |
| F1-score | 92.0 | 100.0 | 100.0 | 93.0 | 98.0 | 97.0 | 86.0 | 96.0 |
| Accuracy | 95.0 | 91.0 | 92.0 | 88.0 | 91.0 | 90.0 | 96.0 | 98.0 |
| **ABC** | | | | | | | | |
| Precision | 92.0 | 100.0 | 96.0 | 94.0 | 100.0 | 95.0 | 93.0 | 97.0 |
| Recall | 91.0 | 100.0 | 100.0 | 95.0 | 93.0 | 76.0 | 100.0 | 100.0 |
| F1-score | 92.0 | 100.0 | 98.0 | 93.0 | 95.0 | 91.0 | 87.0 | 98.0 |
| Accuracy | 93.0 | 90.0 | 92.0 | 86.0 | 93.0 | 91.0 | 95.0 | 96.0 |
| **BSVM** | | | | | | | | |
| Precision | 91.0 | 100.0 | 99.0 | 92.0 | 89.0 | 74.0 | 83.0 | 94.0 |
| Recall | 91.0 | 100.0 | 100.0 | 90.0 | 86.0 | 85.0 | 86.0 | 98.0 |
| F1-score | 93.0 | 100.0 | 100.0 | 91.0 | 86.0 | 77.0 | 87.0 | 95.0 |
| Accuracy | 93.0 | 93.0 | 92.0 | 86.0 | 90.0 | 91.0 | 93.0 | 96.0 |
| **BMLP** | | | | | | | | |
| Precision | 92.0 | 100.0 | 93.0 | 90.0 | 88.0 | 73.0 | 81.0 | 91.0 |
| Recall | 91.0 | 100.0 | 97.0 | 93.0 | 88.0 | 86.0 | 88.0 | 97.0 |
| F1-score | 90.0 | 100.0 | 91.0 | 88.0 | 83.0 | 75.0 | 84.0 | 91.0 |
| Accuracy | 92.0 | 91.0 | 93.0 | 89.0 | 91.0 | 88.0 | 92.0 | 94.0 |

Table 8 shows the results of percentage precision, recall, F1-score and accuracy analysis of SAVEE MFCC1 features. In the results, the maximum percentage precision F1-score of the neutral emotion had achieved 94.0% using the RDF ensemble algorithm. This F1-score was quite high when compared to the F1-score obtained from RAVDESS database using the same features. The percentage increase in overall precision of neutral emotion was observed to be 133.33%, 77.36%, 18.57%, 12.66% and 10.39% using GBM, RDF, BMLP, ABC and BSVM respectively. Moreover, RDF and GBM algorithms achieved perfect prediction (100.0%) for recognizing the surprise emotion. The results show that MFCC features still achieved low recognition rates for the fear emotion as reported by other authors [25]. However, RDF (94.0%) and GBM (91.0%) show higher precision rates for the neutral emotion, which is higher than what was reported in [11].

**Table 8.** Percentage precision, recall and F1-score and accuracy on SAVEE MFCC1.

| | Emotion | | | | | | |
|---|---|---|---|---|---|---|---|
| **Classifier/Measure** | **Angry** | **Disgust** | **Fear** | **Happy** | **Neutral** | **Sad** | **Surprise** |
| **RDF** | | | | | | | |
| Precision | 65.0 | 24.0 | 65.0 | 48.0 | 94.0 | 48.0 | 100.0 |
| Recall | 65.0 | 83.0 | 57.0 | 65.0 | 52.0 | 83.0 | 100.0 |
| F1-score | 65.0 | 37.0 | 60.0 | 55.0 | 67.0 | 61.0 | 100.0 |
| Accuracy | 69.0 | 66.0 | 60.0 | 75.0 | 64.0 | 77.0 | 81.0 |
| **GBM** | | | | | | | |
| Precision | 65.0 | 29.0 | 75.0 | 26.0 | 91.0 | 62.0 | 100.0 |
| Recall | 62.0 | 67.0 | 48.0 | 55.0 | 62.0 | 72.0 | 100.0 |
| F1-score | 63.0 | 40.0 | 59.0 | 35.0 | 74.0 | 67.0 | 100.0 |
| Accuracy | 70.0 | 66.0 | 57.0 | 74.0 | 62.0 | 79.0 | 84.0 |
| **ABC** | | | | | | | |
| Precision | 66.0 | 28.0 | 64.0 | 36.0 | 89.0 | 59.0 | 98.0 |
| Recall | 68.0 | 29.0 | 62.0 | 40.0 | 91.0 | 59.0 | 100.0 |
| F1-score | 65.0 | 28.0 | 63.0 | 38.0 | 91.0 | 58.0 | 99.0 |
| Accuracy | 60.0 | 57.0 | 53.0 | 66.0 | 62.0 | 79.0 | 83.0 |
| **BSVM** | | | | | | | |
| Precision | 65.0 | 23.0 | 61.0 | 41.0 | 85.0 | 57.0 | 98.0 |
| Recall | 64.0 | 26.0 | 57.0 | 38.0 | 81.0 | 61.0 | 100.0 |
| F1-score | 66.0 | 29.0 | 62.0 | 36.0 | 83.0 | 58.0 | 99.0 |
| Accuracy | 56.0 | 54.0 | 51.0 | 69.0 | 56.0 | 77.0 | 85.0 |
| **BMLP** | | | | | | | |
| Precision | 64.0 | 23.0 | 59.0 | 38.0 | 83.0 | 54.0 | 95.0 |
| Recall | 63.0 | 25.0 | 55.0 | 39.0 | 85.0 | 57.0 | 97.0 |
| F1-score | 64.0 | 25.0 | 59.0 | 37.0 | 81.0 | 55.0 | 96.0 |
| Accuracy | 52.0 | 50.0 | 49.0 | 51.0 | 53.0 | 64.0 | 69.0 |

Table 9 shows the analysis results of the percentage precision, recall, F1-score and accuracy obtained after performing the classification with SAVEE MFCC2 features. It can be observed from the table that all the learning algorithms performed well for recognizing the surprise emotion, but performed poorly in recognizing other emotions. In general, all learning algorithms achieved a low prediction of emotions with the exception of surprise emotion in which the perfect F1-score was obtained using the RDF algorithm. The lowest precision value of 93.0% was achieved for the surprise emotion using MFCC2 features with BMLP. These results indicate the unsuitability of MFCC, ZCR, energy and fundamental frequency features for recognizing human emotions. However, RDF and GBM gave higher recognition results for happy and sad emotions, RDF gave a higher recognition result for fear emotion while ABC and BSVM gave higher recognition results for the sad emotion that the traditional and bottleneck features [27].

**Table 9.** Percentage precision, recall and F1-score and accuracy on SAVEE MFCC2.

| | Emotion | | | | | | |
|---|---|---|---|---|---|---|---|
| Classifier/Measure | Angry | Disgust | Fear | Happy | Neutral | Sad | Surprise |
| **RDF** | | | | | | | |
| Precision | 71.0 | 85.0 | 69.0 | 68.0 | 95.0 | 59.0 | 100.0 |
| Recall | 70.0 | 84.0 | 65.0 | 67.0 | 68.0 | 88.0 | 100.0 |
| F1-score | 71.0 | 84.0 | 68.0 | 68.0 | 74.0 | 71.0 | 100.0 |
| Accuracy | 77.0 | 73.0 | 63.0 | 70.0 | 69.0 | 82.0 | 100.0 |
| **GBM** | | | | | | | |
| Precision | 72.0 | 65.0 | 83.0 | 47.0 | 93.0 | 69.0 | 100.0 |
| Recall | 69.0 | 74.0 | 69.0 | 62.0 | 71.0 | 77.0 | 100.0 |
| F1-score | 70.0 | 68.0 | 71.0 | 48.0 | 77.0 | 75.0 | 100.0 |
| Accuracy | 74.0 | 71.0 | 59.0 | 73.0 | 69.0 | 86.0 | 92.0 |
| **ABC** | | | | | | | |
| Precision | 74.0 | 71.0 | 70.0 | 58.0 | 69.0 | 73.0 | 99.0 |
| Recall | 76.0 | 72.0 | 68.0 | 61.0 | 68.0 | 71.0 | 100.0 |
| F1-score | 74.0 | 75.0 | 73.0 | 59.0 | 65.0 | 73.0 | 100.0 |
| Accuracy | 71.0 | 67.0 | 56.0 | 69.0 | 69.0 | 71.0 | 99.0 |
| **BSVM** | | | | | | | |
| Precision | 71.0 | 68.0 | 66.0 | 59.0 | 67.0 | 71.0 | 98.0 |
| Recall | 72.0 | 72.0 | 64.0 | 61.0 | 64.0 | 75.0 | 100.0 |
| F1-score | 70.0 | 70.0 | 66.0 | 60.0 | 65.0 | 73.0 | 100.0 |
| Accuracy | 71.0 | 62.0 | 51.0 | 67.0 | 68.0 | 73.0 | 100.0 |
| **BMLP** | | | | | | | |
| Precision | 70.0 | 66.0 | 64.0 | 57.0 | 66.0 | 67.0 | 93.0 |
| Recall | 72.0 | 70.0 | 66.0 | 59.0 | 68.0 | 72.0 | 97.0 |
| F1-score | 71.0 | 68.0 | 64.0 | 59.0 | 65.0 | 69.0 | 96.0 |
| Accuracy | 56.0 | 52.0 | 49.0 | 65.0 | 54.0 | 64.0 | 85.0 |

Table 10 shows the analysis results of the percentage precision, recall, F1-score and accuracy obtained after performing classification with SAVEE HAF. Accordingly, all the emotions were much easier to recognize using the HAF that had tremendously improved the recognition of the fear emotion reported in the literature to be difficult with lower performance results recorded with traditional and bottleneck features for fear emotion [27]. Moreover, Kerkeni et al. [11] obtained a recognition accuracy rate of 76.16% for the fear emotion using MFCC and MS features based on the Spanish database of seven emotions. These results show that using HAF with ensemble learning is highly promising for recognizing emotions in human speech. The study findings were consistent with the literature that ensemble learning gives a better predictive performance through the fusion of information knowledge of predictions of multiple inducers [75,85]. The ability of ensemble learning algorithms to mimic the nature of human by seeking opinions from several inducers for informed decision distinguish them from inducers [73]. Moreover, it can be seen across Tables 4–10 that the set of HAF presented the most effective acoustic features for speech emotion recognition, while the set MFCC1 presented the worst features. In addition, the study results supported the hypothesis that a combination of acoustic features based on prosodic and spectral reflects the important characteristics of speech [42,53,54].

**Table 10.** Percentage precision, recall and F1-score and accuracy on SAVEE HAF.

| Classifier/Measure | Angry | Disgust | Fear | Happy | Neutral | Sad | Surprise |
|---|---|---|---|---|---|---|---|
| | | | | Emotion | | | |
| **RDF** | | | | | | | |
| Precision | 100.0 | 100.0 | 94.0 | 100.0 | 100.0 | 100.0 | 100.0 |
| Recall | 100.0 | 94.0 | 100.0 | 100.0 | 100.0 | 100.0 | 100.0 |
| F1-score | 100.0 | 100.0 | 100.0 | 100.0 | 100.0 | 97.0 | 97.0 |
| Accuracy | 100.0 | 99.0 | 97.0 | 100.0 | 99.1 | 100.0 | 100.0 |
| **GBM** | | | | | | | |
| Precision | 100.0 | 96.0 | 100.0 | 100.0 | 100.0 | 100.0 | 100.0 |
| Recall | 100.0 | 100.0 | 94.0 | 100.0 | 100.0 | 100.0 | 100.0 |
| F1-score | 100.0 | 98.0 | 97.0 | 100.0 | 100.0 | 100.0 | 100.0 |
| Accuracy | 100.0 | 98.1 | 99.0 | 100.0 | 100.0 | 100.0 | 100.0 |
| **ABC** | | | | | | | |
| Precision | 100.0 | 94.0 | 100.0 | 100.0 | 100.0 | 100.0 | 100.0 |
| Recall | 100.0 | 100.0 | 96.0 | 100.0 | 100.0 | 100.0 | 100.0 |
| F1-score | 100.0 | 94.0 | 91.0 | 100.0 | 100.0 | 100.0 | 100.0 |
| Accuracy | 100.0 | 99.0 | 99.0 | 100.0 | 100.0 | 100.0 | 100.0 |
| **BSVM** | | | | | | | |
| Precision | 100.0 | 93.0 | 99.0 | 99.0 | 100.0 | 100.0 | 100.0 |
| Recall | 100.0 | 96.0 | 98.0 | 99.0 | 100.0 | 100.0 | 100.0 |
| F1-score | 100.0 | 96.0 | 99.0 | 100.0 | 100.0 | 100.0 | 100.0 |
| Accuracy | 100.0 | 94.0 | 96.0 | 96.0 | 100.0 | 100.0 | 100.0 |
| **BMLP** | | | | | | | |
| Precision | 100.0 | 90.0 | 98.0 | 97.0 | 100.0 | 100.0 | 100.0 |
| Recall | 100.0 | 93.0 | 97.0 | 99.0 | 100.0 | 100.0 | 100.0 |
| F1-score | 100.0 | 91.0 | 98.0 | 98.0 | 100.0 | 100.0 | 100.0 |
| Accuracy | 100.0 | 93.0 | 95.0 | 99.0 | 100.0 | 100.0 | 100.0 |

Tables 4–10 show that HAF features had a higher discriminating power because higher performance results were achieved across all emotion classes in the SAVEE and RAVDESS corpora. The performance of each ensemble learning algorithm was generally low when MFCC1 and MFCC2 were applied because the sets of the features had a much lower discriminating power compared to HAF. The recognition results across the experimental databases were different because RAVDESS had more instances than SAVEE. Moreover, SAVEE constitutes male speakers only while RAVDESS is comprised of both male and female speakers. According to the literature, this diversity has an impact on the performance of emotion recognition [35]. The literature review revealed that most emotion recognition models have not been able to recognize the fear emotion with a higher classification accuracy of 98%, but the results in Table 4 show that HAF significantly improved the recognition performance of emotion beyond expectation.

## 5. Conclusions

The automatic recognition of emotion is still an open research because human emotions are highly influenced by quite a number of external factors. The primary contribution of this study is the construction and validation of a set of hybrid acoustic features based on prosodic and spectral

features for improving speech emotion recognition. The constructed acoustic features have made it easy to effectively recognize eight classes of human emotions as seen in this paper. The agglutination of prosodic and spectral features has been demonstrated in this study to yield excellent classification results. The proposed set of acoustic features is highly effective in recognizing all the eight emotions investigated in this study, including the fear emotion that has been reported in the literature to be difficult to classify. The proposed methodology seems to work well because combining certain prosodic and spectral features increases the overall discriminating power of the features hence increasing classification accuracy. This superiority is further underlined by the fact that the same ensemble learning algorithms were used on different feature sets, exposing the performance of each group of features regarding their discriminating power. In addition, we saw that ensemble learning algorithms generally performed well. They are widely judged to improve recognition performance better than a single inducer by decreasing variability and reducing bias. Experimental results show that it was quite difficult to achieve high precisions and accuracies when recognizing the neutral emotion with either pure MFCC features or a combination of MFCC, ZCR, energy and fundamental frequency features that were seen to be effective in recognizing the surprise emotion. However, we provided evidence through intensive experimentation that random decision forest ensemble learning of the proposed hybrid acoustic features was highly effective for speech emotion recognition.

The results of this study were generally fantastic, even though the Gaussian noise was added to the training data. The gradient boosting machine algorithm yielded good results, but the main challenge with the algorithm is that its training time is long. In terms of effectiveness, the random decision forest algorithm is superior to the gradient boosting machine algorithm for speech emotion recognition with the proposed acoustic features. The results obtained in this study were quite stimulating, nevertheless, the main limitation of this study was that the experiments were done on acted speech databases in consonance with the research culture. There can be major differences between working with acted and real data. Moreover, there are new features such as breath and spectrogram not investigated in this study for recognizing speech emotion. In the future, we would like to pursue this direction to determine how the new features will compare with the proposed hybrid acoustic features. In addition, future research will harvest real emotion data for emotion recognition using the Huawei OceanConnect cloud based IoT platform and Narrow Band IoT resources in South Africa Luban workshop at the institution of the authors, which is a partnership project with the Tianjin Vocational Institute in China.

**Author Contributions:** Conceptualization by K.Z. and O.O., Data curation by K.Z. Methodology by K.Z. Writing original draft by K.Z. and O.O. Writing review and editing by O.O., Supervision by O.O. All authors have read and agreed to the published version of the manuscript.

**Funding:** This research received no external funding.

**Conflicts of Interest:** The authors declare no conflict of interest.

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
