# Peer review of "Ensemble Learning of Hybrid Acoustic Features for Speech Emotion Recognition"

_algorithms, doi:10.3390/a13030070_

Round 1

Reviewer 1 Report

The authors presented an ensemble learning of hybrid acoustic features for speech emotion recognition. The contribution of this research is clearly presented in the Introduction and the manuscript is well-written. I could recommend the publication of this paper after the following issues are properly addressed/answered.

Comment. Why the authors have not evaluated their proposed framework against other proposed methodologies presented in Section 2.

Comment. The authors MUST significantly improve the quality of Figures 1-3 (increase dvi). They are blur.

Comment. Since you are performing multi-class classification, I need more evidence of the performances of each algorithm. We must see confusion matrices. Overall performance is a “must” but partial performance for each class is absolutely necessary.  I need to see much more extended discussion of the performance in each dataset.

Regarding the utilized performance metrics. Since -score is a harmonic mean of recall and precision it is wise to use all three of them. I suggest the authors to use -score alone or with sensitivity (recall) and specificity metrics.

To this end, I suggest the authors to include the mean -score, mean sensitivity and mean specificity, along with Accuracy in Table 3.

Comment. Regarding the selected classifiers: Random Decision Forest (RDF) and Gradient Boosting Machine (GBM) constitute ensemble algorithms while Classification and Regression Tree (CART) and Support Vector Machine (SVM) do not. In opinion, it is not fair to compare the performance of single and ensemble classifiers.

Thus, I suggest the authors to evaluate the performance of RDF and GBM against some ensemble algorithm such as Voting, Bagging, AdaBoost utilizing as base learners CART or SVM (see "On ensemble techniques of weight-constrained neural networks." Evolving Systems (2020): 1-13” and "Combining bagging and boosting." International Journal of Computational Intelligence 1.4 (2004): 324-333.”

Comment. I noticed that the authors have not utilized the Artificial Neural Networks. Why?

Some minor comments

  • In Table 1, replace “Result” with “Result (%)
  • Replace “F1-score” with “-score”.
  • In line 399, replace “F1-score is the weighted average that conveys the balance between precision and recall [75]” with “-score constitutes a harmonic mean of precision and recall [Improving the evaluation process of students’ performance utilizing a decision support software." Neural Computing and Applications 31.6 (2019): 1683-1694]”

Author Response

Response to Reviewer 1 Comments

Point 1: Why the authors did not evaluated their proposed framework against other proposed methodologies presented in section 2.

Response: We presented a summary of the performance of our proposed framework performance against the work presented in section 2 as tabulated in Table 1. The comparative analysis of studies that have utilized similar databases SAVEE and RAVDESS has also been included at the end of the discussion of the results in Table 4.

Point 2: The authors must significantly improve the quality of Figures 1-3 (increase div) They are blur.

Response: We have replaced Figures 1 – 3 with high quality figures.

Point 3: Since you are performing multi-class classification, I need more evidence of the performances of each algorithm. We must see confusion matrices. Overall performance is a “must” but partial performance for each class is absolutely necessary. I need to see much more extended discussion of the performance in each dataset.

Regarding the utilized performance metrics. Since score is a harmonic mean of recall and precision, it is wise to use all three of them. I suggest the authors to use score alone or with sensitivity (recall) and specificity metrics.

To this end, I suggest the authors to include the mean score, mean sensitivity and mean specificity along with Accuracy in Table 3.

Response: We have extracted the accuracies of each algorithm from the computed confusion matrixes and added new tables to present these results. We followed this route to provide more evidence as recommended and also to minimize the number of tables because we going to end up having 24 tables describing confusion matrices of each algorithm. Instead, we ended up adding few tables with the required information. Score, recall and precision were all used in the experimental study and these are presented in Tables 5 to 10.

Point 4: Regarding the selected classifiers: RDF and GBM constitute ensemble classifiers while CART and SVM do not. In my opinion, it is not fair to compare the performance of single and ensemble classifiers.

Then I suggest the authors to evaluate the performance of RDF and GBM against some ensemble algorithm such as Voting, Bagging, Adaboost utilizing as base learners CART or SVM.

Response: We replaced CART and SVM with Bagging SVM and Adaboost CART using SVM and CART as the base classifier respectively.

Point 5: I noticed that the authors have not utilized the Artificial Neural Networks. Why? In line 399, replace “F1-score is the weighted average that conveys the balance between precision and recall [75]” with “-score constitute a harmonic mean of precision and recall [Improving the evaluation process of students’ performance utilizing a decision support software.” Neural computing and Applications 31.6 (2019): 1683-1694]”

Response: We added an MLP based boosting ensemble where we used MLP as the base classifier. We made the recommended change with regards to F1-score and cited the recommended references.

Reviewer 2 Report

The paper is good at presenting state of the art methods in emotion recognition. However the conclusions and the justifications of the results are somehow vague. It is not understood or detailed why the redundancy of somehow similar features such as MFCC, LPCC plus FFT renders better solution than a single spectral feature.

Minor revisions

  • In page 3, Interspeech 2010 features are mentioned twice  (line 112 and 131). To my knowledge there are not such kind of features
  • In page 4, the sentence "applying PCA to select important features" is inaccurate. PCA does not select features but TRANSFORM them to a different coordinate base
  • In page 4, line 163, You mention Interspeech 2009 standard features. There is not such a set of features to my knowledge
  • In page 4, line 168, it is mentioned "Fourier parameter". It is not clear. Do you mean  Fourier transform?
  •  In Table 1.A, The column “Signal” could be deleted since it is the same for all of them. However, it would be interesting to add a new column, separating the feature set from the classification method. Also it would be very informative to add a new column with “type of recordings, normal speech versus acted speech”
  • In Fig. 1 and the heading of paragraph 3.4 There is a block named “Feature selection” but there is not any algorithm or method mentioned in the paper for doing this. It also mentions “Feature classification”. It would be more appropriate to rewrite simply as “classification” since the features themselves are not classified.
  • In page 9 and  in Table 2, you mention ZCR as prosodic features.  ZCR are features related to acoustics (voiced/unvoicev decision) but not to prosody. Duration characteristic in prosody means phone/word/sentence/rhythm but not ZCR.
  • In Section 4, you mention “data cleaning”. What do you mean by that?
  • In Section 4 you mention “Noise added”, what kind of noise?
  • In Table 3, results should give also confidence factors for each result since the number of examples in the database are very small.
  • The only published data with the same database is given in reference [48]. Please discuss the differences from your results and the published results

Author Response

Response to Reviewer 2 Comments

Point 1: In page 3, Interspeech 2010 features are mentioned twice (line 112 and 131). To my knowledge there are not such features.

Response: This error was corrected, GCZCMT features were used in line 112. Thank you for locating this error with eagle eyes.

Point 2: In page 4, the sentence “applying PCA to select important features” is inaccurate. PCA does not select features but TRANSFORM them to a different coordinate base.

Response: The sentence was corrected as recommended.

Point 3: In page 4 line 163, You mentioned Interspeech 2009 standard features. There is not such set of features to my knowledge.

Response: The statement was corrected as recommended.

Point 4: In page 4, line 168, it is mentioned: Fourier parameter”. It is not clear. Do you mean Fourier transform?

Response: These are a set of harmonic sequences used to detect the perceptual content of voice quality features. These are derived from discrete Fourier transform (DFT). Therefore, they are described as Fourier parameters by people who authored the paper. This is evidence from the title of the paper listed below.

Wang, K.; An, N.; Li, B.N.; Zhang, Y.; Li, L. Speech emotion recognition using Fourier parameters. IEEE Trans. Affect. Comput. 2015, 6, 69–75.

Point 5:  In Table 1A, The column “Signal” could be deleted since it is the same for all of them. However, it would be interesting to add a new column, separating the feature set from the classification method. Also, it would be informative to add a new column with “ type of recordings, normal speech vs acted speech.”

Response: The recommended changes were done.

Point 6: In Fig.1 and the heading of paragraph 3.4 there is a block named “Feature selection” but there is not any algorithm or method mentioned in the paper for doing this. It also mentions “Feature classification”. It would be more appropriate to write simply as “classification” since the features themselves are not classified.

Response: The recommendations were effected

Point 7: In page 9 and in Table 2, you mention ZCR as prosodic features. ZCR are features related to acoustics (voiced/unvoicedv decision) but not to prosody. Duration characteristic in prosody meana phone/word/sentence/rhythm but not ZCR.

Response: The reason we have classified ZCR under the prosody group is because Schuller, Balas et al. and  Pervaiz & Ahmed classified it in the same category. In his book (page 311) entitled “Intelligent audio analysis”, Schuller puts ZCR in the prosody group of features. In the same vein, Balas et al also classify ZCR as a prosodic feature (page 344). Pervaiz & Ahmed also classified ZCR under the prosody category in their paper entitled “Emotion Recognition from Speech using Prosodic and Linguistic Features”. The references of the aforementioned material is given below.

[1]         Balas VE, Kumar A, Ahmed B. Cognitive Informatics and Soft. 2017.

[2]         Schuller BW. Intelligent Audio Analysis. 2013. https://doi.org/10.1007/978-3-642-36806-6.

[3]         Pervaiz M, Ahmed T. Emotion Recognition from Speech using Prosodic and Linguistic Features. Int J Adv Comput Sci Appl 2016;7:84–90. https://doi.org/10.14569/ijacsa.2016.070813.

In fact, we have made references to other research articles [49, 59, 60] in the last two statements of section 3.2.1 in the text.

Point 8: In section 4 you mention “data cleaning” what do you mean by that?

Response: we have deleted the statement because feature extraction was accomplished by the use of a specialized software.

Point 9: In section 4 you mention “Noise added”, what do you mean by that?

Response: We added statistical Gaussian noise to the training data to provide a regularizing effect, reduce overfitting, increase resilience and improve generalization performance of the learning algorithms. Gaussian noise is statistical noise having a probability density function (PDF) equal to that of the normal distribution.

Point 10: In Table 3, results should give also confidence factors for each result since the number of examples in the database are very small.

Response: We performed a confident factor test and added the computed values onto the accuracy result tables in response to point 10.

Point 11: The only published data with the same database is given in reference [48]. Please discuss the differences from your results and the published results.

Response: We have done so by discussing the comparative analysis of studies that have utilized similar databases SAVEE and RAVDESS at the end of the discussion of the results in Table 4.

Reviewer 3 Report

The authors should describe with pseudocode the whole algorithm.
A statistical test should be used for the comparison of the examined methods.
The authors should explain why the proposed methodology seems to work well and present some information about the time efficiency of their method.

Author Response

Response to Reviewer 3 Comments

Point 1: The authors should describe the pseudocode the whole algorithm.

Response: The pseudocode of the whole algorithm in form of a flow chart has replaced Figure 1 as Figure 3 in the text.

Point 2: A statistical test should be used for the comparison of the examined methods.

Response: We performed a confident interval test and added the computed values to Table 4 for each corpus in response to point 2.

Point 3: The authors should explain why the proposed methodology seems to work well and present some information about the time efficiency of their method.

Response: The proposed methodology seems to work well because combining certain prosodic and spectral features increases the overall discriminating power of the features, hence increasing classification performance in terms of accuracy, precision, recall and F1-score. This superiority is further underlined by the fact that the same algorithms were used on different feature sets, exposing the performance of each group of features regarding their discriminating power. Table 3 containing the computational; training time has been added to the paper in the results section.

Round 2

Reviewer 1 Report

The authors have somehow addressed the previous comment.

Some minor suggestions

  • Replace "score" with "F1-score"
  • Replace the works citing Bagging and AdaBoost ensemble models with more recent works.

Author Response

Reviewer 1

Some minor suggestions

  • Replace "score" with "F1-score"
  • Replace the works citing Bagging and AdaBoost ensemble models with more recent works.

Response 1

Done

Response 2

Additional references are;

Yaman, E., Subasi, A. Comparison of bagging and boosting ensemble machine learning methods for automated EMG signal classification. BioMed Research International, 2019, 2019.

Navarro, C.F., Perez, C.A, Color–texture pattern classification using global–local feature extraction, an SVM classifier with bagging ensemble post-processing. Applied Sciences, 9(15), 2019, 3130.

Wu, Y., Ke, Y., Chen, Z., Liang, S., Zhao, H., Hong, H., Application of alternating decision tree with AdaBoost and bagging ensembles for landslide susceptibility mapping. Catena, 187, 2020, 104396.

Xing, H.J., Liu, W.T. Robust AdaBoost based ensemble of one-class support vector machines. Information Fusion, 55, 2020, 45-58.

Reviewer 3 Report

The paper could be accepted in the current form

Author Response

Reviewer 3

The paper could be accepted in the current form. Thanks.

Round 3

Reviewer 1 Report

I recommend the publication of the manuscript